# Evaluation of Health-Related Quality of Life in Patients with Euthyroid Hashimoto’s Thyroiditis under Long-Term Levothyroxine Therapy: A Prospective Case-Control Study

**DOI:** 10.3390/jcm13113082

**Published:** 2024-05-24

**Authors:** Nilgun Tan Tabakoglu, Mehmet Celik

**Affiliations:** 1Hospital Health Research and Development Center, Faculty of Medicine, Trakya University, Edirne 22030, Turkey; 2Department of Internal Medicine—Endocrinology and Metabolism, Trakya University, Edirne 22030, Turkey; drmehmetcelik@hotmail.com

**Keywords:** Hashimoto’s thyroiditis, SF-12, quality of life, hypothyroidism

## Abstract

**Objectives**: To investigate quality of life using the SF-12 scale in euthyroid Hashimoto’s thyroiditis patients on levothyroxine therapy for at least three years. **Methods**: This prospective case–control study included 44 euthyroid Hashimoto’s thyroiditis patients and 44 matched controls, conducted at a university hospital’s endocrinology clinic from 6 November to 30 December 2023. Participants completed the SF-12 questionnaire; data were analyzed using Shapiro–Wilk, Student’s *t*-test, Mann–Whitney U, Yates chi-squared, and Spearman’s tests. **Results**: The study involved 88 participants (Hashimoto’s group: 35 females, 9 males; control group: 31 females, 13 males), with average ages of 49.50 and 47.43 years old, respectively. Significant differences were observed in TSH, T4 levels, and family history (*p* < 0.05). The Hashimoto’s thyroiditis group showed higher thyroid peroxidase antibodies (95.69 IU/mL) and lower scores on both physical and mental sub-dimensions of SF-12, with a significant difference in physical scores (*p* < 0.05). Significant correlations were found between age and Anti-TG; Anti-TPO and Anti-TG; BMI and T3; TSH and T4; HDL and triglycerides; MCS-12 and PCS-12; Anti-TPO and T3; cholesterol and T3; and LDL and cholesterol (*p* < 0.05). Other variables showed no significant correlations (*p* > 0.05). **Conclusions:** Our study shows that effective control of hypothyroidism is not sufficient to reduce the negative effects of Hashimoto’s thyroiditis on patients’ health-related quality of life. Beyond the normalization of hormone levels, comprehensive therapeutic strategies targeting the autoimmune aspects of the disease are essential for the management of Hashimoto’s thyroiditis. This study provides a foundation for developing effective therapies that can enhance quality of life for patients with Hashimoto’s thyroiditis.

## 1. Introduction

Hypothyroidism is a clinical condition characterized by a lack of thyroid hormones in body tissues or decreased activity of these hormones, resulting in a slowed metabolism. The most common cause of hypothyroidism in the world is inadequate iodine intake. In regions with adequate dietary iodine intake, the most common cause is ‘chronic autoimmune thyroiditis’, also known as Hashimoto’s thyroiditis (HT) [1,2]. HT is the most common autoimmune thyroid disease characterized by thyroid gland enlargement with atrophy secondary to autoimmune-induced inflammation [3].

The prevalence of HT in the general population ranges from 0.3 to 1.5 per thousand, with a 5 to 20-times-higher prevalence in the female population than in the male population [4].

Since hypothyroidism presents with non-specific clinical symptoms, the diagnosis is usually based on biochemical tests. Serum thyroid stimulating hormone (TSH) and free thyroxine (T4) levels are used. In overt primary hypothyroidism, serum TSH levels are high and free T4 levels are low. Another condition in which serum TSH levels are high but free T4 levels remain within the reference range is subclinical hypothyroidism [5]. In subclinical hypothyroidism, patients are often detected without the typical clinical signs of hypothyroidism and usually during routine health check-ups. The prevalence of subclinical hypothyroidism ranges from 5% to 15% and increases with age. In population surveys, it has been found to be approximately 4.4% in men and 7.5% to 8.5% in women. Subclinical hypothyroidism is more common in older women, with a prevalence ranging from 7% to 18% and 2% to 15% in men. [6]. Approximately 4% to 5% of HT patients with subclinical hypothyroidism are diagnosed with overt hypothyroidism each year [1].

The primary hormone secreted by the thyroid gland is T4. Small amounts of triiodothyronine (T3) are also secreted [7]. Thyroid hormones exert their effects by activating the transcription of many genes at the nuclear level and increasing the functional capacity of many cells they affect. However, T4 cannot bind efficiently to thyroid hormone receptors in other systems in the body [8]. Therefore, to initiate cellular activity, T4 must be converted to T3, which binds efficiently to thyroid hormone receptors. The more biologically active T3 is produced in peripheral tissues by the deiodination of T4 by type-1 and -2 iodothyronine deiodinase enzymes; thus, T3 is ready to play a fundamental role in hemostasis in the body [9].

Thyroid hormones regulate normal growth, development, and various homeostatic functions, particularly the production of energy and heat. In individuals diagnosed with hypothyroidism, the lack of these hormones or their ineffective action in target organs can result in a wide range of clinical symptoms that can significantly reduce overall health and quality of life. These symptoms include fatigue, weakness, increased body weight, distraction, memory problems, hypersensitivity to cold, hair loss, thickening of the voice, constipation, dry skin, menstrual irregularities, and depression. The clinical manifestations of hypothyroidism vary depending on the etiology, duration, and severity of the disease. It can affect all organs and systems of the body [1,10].

The diagnosis of HT is based on clinical, laboratory, and ultrasonographic findings. In addition to the presence of thyroid peroxidase antibody (Anti-TPO) and thyroid globulin antibody (Anti-TG) positivity in patients with hypothyroid symptoms, thyroid ultrasonography is diagnosed with decreased echogenicity, parenchymal heterogeneity, hypervascularization, increased volume, and micronodules due to lymphocytic infiltration [11].

Currently, levothyroxine (LT4) is commonly used for the treatment of hypothyroidism resulting from HT. The long half-life of this drug, up to one week, makes it possible for patients to take a single daily dose, thus simplifying the treatment process. LT4 is also largely converted to T3 in peripheral tissues, helping to maintain stable levels of the hormone T3 in the body. This process plays a critical role in controlling symptoms of hypothyroidism and maintaining a normal thyroid function [12,13]. It is essential to monitor TSH levels when starting treatment and during follow-up. The general consensus is that all patients with a serum TSH level of 10 mU/L should be treated. The goal of treatment is to achieve the set TSH target (0.4–4.0 mIU/L) [14]. However, when adjusting treatment according to the TSH level, it is essential to remember that TSH release increases with age, obesity, and smoking may slightly increase TSH levels, pregnancy may decrease TSH levels, and TSH release has a daily rhythm [5]. These factors should be considered in treatment planning and the interpretation of TSH levels.

Recently, a health approach that emphasizes improving patients’ quality of life is as important as treating chronic diseases has been adopted [15,16,17].

The health-related quality of life scale (SF-12) refers to a wide range of health-related factors, such as an individual’s physical condition, mental health, and social relationships. It is a critical assessment tool often used to measure the success of health services. This assessment tool has two main components: the summary of physical status (PCS-12) and the summary of mental status (MCS-12) [18]. As with other chronic diseases, untreated hypothyroidism and associated HT reduce SF-12 values.

This study is important because HT is one of the most common thyroid diseases, affecting 5% of the population, and its prevalence is increasing gradually [19,20]. Furthermore, Anti-TPO and Anti-TG levels remain elevated in patients on long-term LT4 therapy. This suggests that autoimmunity is still active. This is one of the main reasons for the difficulty in managing the treatment of HT patients, because the treatments these patients receive are directed at the symptoms of hypothyroidism caused by the disease. However, they do not target the root cause, which is autoimmunity. This may explain why, despite being treated for hypothyroidism, patients’ quality of life was not as high as that of healthy individuals.

This study aimed to compare the SF-12s of euthyroid Hashimoto’s thyroiditis patients who had been on LT4 treatment for at least 3 years and healthy volunteers with equivalent demographic characteristics who did not have chronic diseases that reduce quality of life.

It is generally accepted that the decrease in the quality of life in HT patients is due to hypothyroidism. The fact that this study was conducted with a group of euthyroid patients who received long-term hypothyroidism treatment and had no comorbidities is very significant in terms of evaluating the effect of HT on quality of life independently of hypothyroidism. At the same time, this study may contribute to the development of treatment strategies to improve the quality of life of HT patients and encourage new research in this field.

## 2. Materials and Methods

### 2.1. Study Design and Participants

This is a prospective case–control study conducted with HT patients who applied to the endocrinology outpatient clinic of our hospital between 6 November and 30 December 2023, and healthy volunteers who agreed to participate in the study. All participants were euthyroid. The case group consisted of 44 patients receiving LT4 treatment with a diagnosis of HT; the control group consisted of 44 healthy volunteers without thyroid disease or chronic diseases (diabetes mellitus, heart failure, chronic kidney disease, or autoimmune diseases) that reduce the quality of life.

The inclusion criteria of the case group in the study were as follows: being 18 years of age or older, being followed for at least 3 years with a diagnosis of HT and being under LT4 treatment, having TSH levels within the euthyroid reference range (TSH levels of 0.4–5.0 mU/L in young, middle-aged individuals; upper limit of 6 mU/L for individuals aged 65–70 years old; upper limit of 7.5 mU/L for individuals aged 80 years and older) at the time of participation in the study, agreeing to participate in the study, and signing the consent form [1,5].

Exclusion criteria: Age younger than 18 years; diagnosis of secondary, tertiary, or subclinical hypothyroidism; possible pregnancy or pregnancy during the study; breastfeeding during the study; TSH levels outside the accepted euthyroid reference ranges; diagnosis of neuropsychiatric disease; being diagnosed with an endocrinologic disease other than thyroid; being diagnosed with a severe disease other than thyroid (heart failure, chronic kidney disease, cancer, hypertension, or autoimmune diseases) that affects quality of life; and being on medication (chemotherapy) that may affect TSH, T3, or T4 levels.

The control group consisted of male and female volunteers aged 18 years and over whose health status was verified using the e-nabız system in our country, who did not have any chronic disease, who were not pregnant or likely to become pregnant, for female volunteers, who were not breastfeeding, and who had laboratory data on T3, T4, and TSH levels in routine health checks performed within the last year [21].

Participants who agreed to participate in the study and met the inclusion criteria were asked to sign an informed consent form, and the researcher administered the questionnaire using a face-to-face interview method. In the first part of the questionnaire, questions were asked to the participants to determine demographic characteristics (age and gender), health status (smoking and alcohol use, duration of hypothyroidism diagnosis, diet, medication use, comorbidities, and whether they had undergone thyroid surgery) and the answers were recorded. In the continuation of the questionnaire, the SF-12 scale was applied to determine the participants’ quality of life. After obtaining permission from the central directorate of our hospital to access the results of tests and examinations by protecting the personal data of the patients, the examinations requested from the participants during the routine follow-up of the case group and from the healthy volunteers at any time in the last year due to routine health examinations were found and recorded from the hospital automation system.

In our study, the inclusion criterion for the HT group was that they had received LT4 treatment for at least three years. This criterion was set in order to examine the effects on the quality of life of patients on long-term treatment in more detail. However, although we set the lower limit of the follow-up period, the upper limit was not questioned or recorded. This resulted in the inability to calculate median or mean values for the study’s follow-up period. This methodological choice should be considered among the potential limitations of the study. Furthermore, variations in the duration of LT4 use were not identified as a central variable to test the study’s main hypothesis.

### 2.2. SF-12 Questionnaire Form

The SF-12 questionnaire was developed to assess individuals’ health-related quality of life. This tool examines the general well-being of people older than 14 years of age in two different dimensions through physical and mental state summaries (PCS and MCS). The questionnaire consists of 12 questions with various Likert scales and a yes–no style. The resulting scores are standardized and transformed for the final analysis [22]. The researchers who developed the questionnaire reported in a study that the United States norms were valid in evaluating the scores and that the mean and standard deviation should be 50 ± 10 [23]. The validity and reliability study of the Turkish translation of this questionnaire form has been conducted, and it is stated that the measurement tool has reached sufficient reliability values. [24]. ‘How to Score the SF-12 Physical and Mental Health Summary Scales’ was used to calculate the SF-12, PCS-12, and MCS-12 scores, and the results were checked with the orthotoolkit online calculation tool [25,26]. Please see the Appendix A for details of the SF-12 questionnaire.

### 2.3. Ethical Approval

Ethics committee approval for this study was obtained from the Ethics Committee of the Faculty of Medicine of our University with the decision number and protocol code TÜTF-GÖBAEK 2023/394 dated 13 November 2023, and all procedures in this study were performed under the Declaration of Helsinki and its subsequent amendments.

### 2.4. Statistical Analysis

In order to test a difference of 0.618 effect size calculated between hypothyroidism and control groups based on the SF-12 quality of life scores in Romero-Gómez et al. study with a 5% margin of error and 80% power, 42 patients from each group should be included in the study [22]. Taking into account possible missing data, the study was completed with 44 HT patients for the Hashimoto’s thyroiditis group and 44 healthy volunteers for the control group.

Descriptive statistics (mean, standard deviation, and number, %) regarding demographic and clinical characteristics in HT thyroiditis and control groups and Anti-TPO and Anti-TG in HT groups were presented. The conformity of quantitative data to the normal distribution in the HT thyroiditis and control groups was analyzed by the Shapiro–Wilk test. Student’s *t*-test was used to compare quantitative data with normal distribution (age, BMI, Hg, and cholesterol) between the groups, and Mann–Whitney U-test was used to compare quantitative data that did not conform to normal distribution (alcohol—year, smoking—packets per year, TSH, T3, T4, triglycerides, HDL, LDL, PCS-12, and MCS-12). Yates chi-squared test was used to compare categorical data (gender, alcohol use, smoking, family history, CAD, and anti-lipidemic drug use) between groups. Spearman’s correlation analysis was used to examine relationships between factors (age, anti-TPO, anti-TG, BMI, TSH, T3, T4, triglycerides, cholesterol, HDL, LDL, PCS-12, and MCS-12) in the HT group. PCS-12 and MCS-12 of the SF-12 scale were calculated in the HT thyroiditis and control groups, and are shown graphically with a box plot. *p* < 0.05 was accepted as the limit of statistical significance. All data were statistically analyzed using the SPSS (IBM SPSS Statistics for Windows, Version 20.0, IBM Corp., Armonk, NY, USA) database.

## 3. Results

This study was conducted with 44 HT groups and 44 control groups consisting of people similar in terms of age, gender, BMI, and habits, while both groups were compared in terms of clinical characteristics and the SF-12 scale. A comparison of demographic and clinical characteristics between the groups is shown in Table 1. A statistically significant difference was found between the groups regarding TSH, T4 levels, and family history (*p* < 0.05). There was no significant difference between the groups regarding other parameters (*p* > 0.05).

The mean Anti-TPO and Anti-TG levels in the HT group are shown in Table 2.

The mean values of Anti-TPO and Anti-TG antibody levels in the HT group are shown in Table 2. As a result of the analysis of 44 patients in the HT group, the mean value of Anti-TPO antibodies was found to be 95.69 IU/mL, which exceeds the standard limit of <34 IU/mL. The mean Anti-TG antibody level was 38.28 IU/mL, which is close to the upper limit of the normal range of <40 IU/mL.

The relationships between variables in the HT group were analyzed using the two-way Spearman’s correlation test, and the results are shown in Table 3.

There was a negative correlation between age and Anti-TG (r = −0.486, *p* = 0.001) and a positive correlation between Anti-TPO and Anti-TG (r = 0.446, *p* = 0.002); there was also a positive correlation between BMI and T3 (r = 0.341, *p* = 0.024) and a negative correlation between TSH and T4 (r = −0.556, *p* = 0.000). HDL levels were negatively correlated with triglyceride levels (r = −0.392, *p* = 0.008), while MCS-12 and PCS-12 were positively correlated (r = 0.302, *p* = 0.047). There was a positive correlation between Anti-TPO and T3 (r = 0.311, *p* = 0.040) and a negative correlation between cholesterol and T3 (r = −0.303, *p* = 0.046). A positive correlation was observed between LDL and cholesterol (r = 0.945, *p* = 0.000).

No statistically significant correlations were found in the relationships between other variables.

Statistical comparisons of PCS-12 and MCS-12 sub-dimension scores of the SF-12 scale between the groups are shown in Table 4.

In addition, the distribution of PCS-12 scores between the groups is shown in Figure 1, and the distribution of MCS-12 scores is shown in Figure 2 with box plots. PCS-12 and MCS-12 scores were lower in the HT group than in the control group. While there was a statistically significant difference between the groups in terms of PCS-12 scores (PCS-12, *p* = 0.049), there was no statistically significant difference in terms of MCS-12 scores (MCS-12, *p* = 0.346).

## 4. Discussion

In this study, we compared SF-12 quality of life scores between euthyroid HT patients who had been on LT4 treatment for at least three years and had no other chronic disease that could affect quality of life and a control group of healthy volunteers. Our study shows that the quality of life scores of HT patients continue to remain low and that the role of hypothyroidism in disease-related symptoms is only one factor, while the main reason may be that the systems that are effective in the development of HT remain active.

HT is a chronic disease that mostly affects women, causes both psychological and physiological symptoms, and negatively affects the quality of life of diagnosed individuals [10]. Although euthyroidism is achieved in patients with HT under medical treatment, there is no biochemical test or measurement to describe how the patient feels physically and mentally.

In current research, SF-12 is a widely used scale to measure personal quality of life. We aimed to investigate the quality of life of euthyroid HT patients under treatment using this scale.

In a study investigating the relationship of HT with age and gender, it was found that patients aged 40–50 years old constituted 46% of the entire group, and 82% of the HT group were women in the same study [27]. In 2022, a meta-analysis of 48 studies from different parts of the world investigated the epidemiology of HT in adults and showed that the prevalence of HT in women was 3.86-times higher than in men [28]. Another study investigating the diagnosis and treatment of HT mentioned that genetic predisposition and X chromosome inactivation play a role in the development of HT and explained both the reason for the higher prevalence in women and that those with a positive family history are more at risk than those with a negative family history [29]. The findings in our study are consistent with this literature. The mean age of our HT group was 49.50 ± 12.09 years old, and 79.5% were female, which is 3.88-times higher than that of male patients. In addition, the number of patients with a positive family history in the HT group was 26, constituting 59.1% of the group. The difference with the control group was statistically significant (*p* = 0.002; Table 1).

Metabolic syndrome is known to be more common in HT patients [30]. Because the dominant thyroid receptor in the liver is thyroid receptor alpha (TRα). TSH regulates lipid metabolism in hepatocytes through this receptor. It also inhibits hepatic bile acid synthesis via the hepatocyte nuclear factor 4 (HNF4)-CYP7A1 signaling pathway and reduces cholesterol production by inhibiting hydroxymethylglutaryl-CoA reductase (HMGCR). A decrease in thyroid hormone levels causes a decrease in the number of glucose receptors in the beta cells of the pancreas, leading to lipolysis and free fatty acid synthesis in the liver. In hypothyroidism, low-density lipoprotein (LDL) receptors are reduced in hepatic cells, and cholesterol absorption from the intestine is increased, leading to inefficient LDL metabolism. Patients with hypothyroidism-induced hepatosteatosis are highly susceptible to oxidative stress due to mitochondrial dysfunction [31]. This leads to chronic comorbidities (hypertension, diabetes mellitus, and coronary artery disease) and, thus, to a decrease in the quality of life of patients. The findings in our study are consistent with the literature [32]. Triglycerides, HDL, LDL, and cholesterol levels were lower in our HT group compared to the control group. The reason for this is the anti-lipidemic and anti-triglyceridemic drugs used by these patients due to metabolic syndrome. Because in our study, the use of anti-lipidemic and anti-triglyceridemic drugs was higher in the HT group compared to the control group. At the same time, coronary artery disease, which is also a component of the metabolic syndrome, was more common in the HT group compared to the control group. All these differences were not statistically significant (*p* > 0.05; Table 1). This may be explained by the fact that the HT group was followed up for 3 years and received an appropriate dose of LT4 treatment to bring TSH into the normal range. Appropriate doses of medical treatment and dietary changes are thought to slow the inflammatory process, leading to a decrease in Anti-TPO and Anti-TG levels to levels lower than those at the initial diagnosis, improvement in thyroid function, and regression in metabolic disorders [30].

In another study comparing 81 HT patients receiving LT4 treatment with 54 healthy volunteers, TSH and T4 levels were significantly higher in the HT group compared to the control group [33]. The results of our study are consistent with the literature (TSH, *p* = 0.003; T4, *p* = 0.021), as shown in Table 1. This is because T3 and T4 production is much lower in patients with HT than it should be due to atrophic thyroid tissue, so TSH levels are typically elevated to increase hormone production. Appropriate LT4 therapy aims to return serum TSH levels to the normal range. The T3 requirement of patients with HT is met by the conversion of LT4 to T3. Therefore, serum TSH and T4 levels may be higher, while T3 levels may be lower in patients with HT. In our study, T3 levels were lower in patients with HT compared to the control group. However. This difference was not statistically significant (*p* = 0.054; Table 1).

Health-related quality of life allows us to understand how a person is affected by illness or treatment processes and how overall health status is reflected in various aspects of life. This definition is frequently encountered in chronic diseases and treatment processes. HT is a chronic disease that affects patients both physically and mentally and reduces their quality of life [34,35]. In our study, we found that patients with HT under LT4 treatment for at least three years had lower SF-12 mental and physical sub-scores compared to the control group; the difference in the physical sub-score was statistically significant, but the decrease in the mental sub-score was not (PCS-12, *p* = 0.049; MCS-12, *p* = 0.346; Table 4).

SF-12 may be affected by sociodemographic conditions, such as marital status, education level, and monthly income level. This is one of the limitations of our study. Although our study did not include this information about the patients, a study conducted in 2020 with 152 hypothyroid patients treated with LT4 and 238 controls showed that sociodemographic data did not affect SF-12 results. In the same study, age and BMI were found to have negative effects on quality of life. However, even after isolating the effects of these two variables, it was observed that hypothyroidism negatively affected quality of life. According to this study, MCS-12 and PCS-12 sub-scores were statistically significantly lower compared to the control group [22]. This result is compatible with our study. However, although the MCS-12 sub-score was lower in our study compared to the control group, no statistically significant difference was found. The reason for this may be that only women were included in the above study regardless of the etiology of hypothyroidism, because the case group in our study consisted of both men and women and only HT patients. Another reason may be the small number of cases in our study. In another study, patients with HT receiving LT4 treatment were re-evaluated for quality of life after 6 months, and despite clinical improvement, they still had worse scores compared to the control group. However, they only found a statistically significant decrease in the MCS-12 result [36]. This result is also consistent with our study. However, the decrease in MCS-12 scores was not statistically significant in our study, but it was still lower than in the control group. The reason for this is that the data at the time of diagnosis and 6 months later were compared in the literature. In our study, HT patients who received LT4 treatment for at least 3 years were compared with the control group. In addition, in another study investigating the quality of life in 84 patients with HT, it was reported that high Anti-TPO levels were associated with an increased risk of depression, even in euthyroid patients [37]. In our study, HT patients’ Anti-TPO levels were approximately three-times higher than the normal range (Table 2). This explains why the MCS-12 scores of the HT group in our study were lower than those of the control group, although not statistically significant. (Table 4; *p* = 0.346). In a study similar to ours, 44 untreated HT patients, 49 treated euthyroid HT patients, and a control group of 31 patients were compared in terms of the effect of thyroid autoimmunity on mental status in euthyroid Hashimoto’s thyroiditis. Using SF-36, Beck depression, and Beck anxiety questionnaires, the study pointed out that the low MCS-12 scores of euthyroid HT patients may be due to thyroid autoimmunity independent of LT4 replacement therapy, which is a similar result to our study [38].

Another study similar to ours revealed that Anti-TPO levels were significantly associated with symptom burden and quality of life in 426 euthyroid female patients with HT [39]. The results of our study are consistent with the current study, showing that patients with HT still had Anti-TPO levels above normal limits despite long-term treatment and had lower SF-12 quality of life scores compared to the control group.

Another study on patients with Hashimoto’s thyroiditis showed that LT4 treatment not only normalizes serum thyroid hormone levels, but may also have significant effects on brain function. In particular, the literature reviewed in 2021 highlights the potential for LT4 to fail to provide adequate levels of T3 hormone in brain tissue [39]. This may explain why some patients do not experience the expected improvement in quality of life despite treatment. Researchers have noted that the local regulation of T3 levels in brain cells, particularly through deiodinases type 2 and type 3, may not be fully achieved with LT4 monotherapy [40]. This may be the reason why cognitive functions and quality of life scores measured by the SF-12 questionnaire were lower in HT patients treated with LT4 compared to the control group in our study.

The results of the Spearman’s correlation analysis in Table 3 illuminate some crucial relationships in the HT group. In particular, a significant negative correlation was found between age and Anti-TG (r = −0.486, *p* = 0.001), suggesting that as age increases, Anti-TG levels decrease. Similarly, a significant positive correlation was found between Anti-TPO and Anti-TG (r = 0.446, *p* = 0.002), indicating that, as Anti-TPO levels increase, Anti-TG levels tend to increase (Table 3). This suggests that the thyroid status observed in HT patients is related to the autoimmune system and plays an essential role in the pathology of the disease.

Furthermore, the relationship between BMI and T3 was also significant (r = 0.341, *p* = 0.024), indicating that increases in BMI are associated with increases in T3 levels. In addition, TSH level was significantly negatively correlated with T4 (r = −0.556, *p* = 0.000), reflecting this expected inverse relationship in thyroid function dynamics (Table 3).

HDL levels were found to have a strong negative correlation with triglyceride levels (r = −0.392, *p* = 0.008), reflecting the typical inverse relationship between these two lipid profile components. Finally, the mental component summary, MCS-12, showed a significant positive correlation with PCS-12 (r = 0.302, *p* = 0.047), suggesting that improvements or declines in physical health scores are associated with similar changes in mental health scores (Table 3).

These significant correlations highlight the complex interplay between thyroid function, lipid profiles, and overall quality of life.

In addition, a positive correlation was found between Anti-TPO and T3 (r = 0.311, *p* = 0.040), indicating that T3 levels increase with increasing Anti-TPO levels, and thyroid autoimmunity may affect thyroid hormone levels. Furthermore, a negative correlation (r = −0.303, *p* = 0.046) was found between cholesterol and T3, indicating that high cholesterol levels may negatively affect T3 levels. A very high and significant positive correlation was observed between LDL and cholesterol (r = 0.945, *p* = 0.000), indicating that cholesterol has a strong relationship with LDL levels within lipid metabolism, which is important for cardiovascular risk assessments (Table 3). These findings make important contributions to better understanding the interactions between thyroid function and lipid profile, and clinical practice should consider the potential impact of these interactions on patient management.

On the other hand, in a study investigating the effects of autoimmunity on quality of life in 84 euthyroid HT patients, it was found that high Anti-TPO and Anti-TG antibody levels were significantly negatively correlated with quality of life scores and patients with high Anti-TPO and Anti-TG levels had significantly lower quality of life scores [37]. In our study, unlike this study, no significant correlation was found between Anti-TPO and Anti-TG and PCS-12 and MCS-12. Our study shows that these biomarkers do not have a direct effect on patients’ quality of life (*p* > 0.05; Table 3). This may be due to the small sample size, which is one of the limitations of our study. However, our results are in line with the same study in that our HT group had high thyroid autoantibodies and low quality of life scores despite long-term treatment. These results suggest that the clinical management of HT and patient care strategies should be considered in depth and may play a critical role in understanding how hormonal imbalances and immune reactivity affect the course of the disease.

Our results show that, despite achieving euthyroid status with long-term LT4 treatment, HT patients continue to live with poor quality of life regarding mental and especially physical health. These findings are in line with recent research. This may be attributed to underlying mechanisms, such as increased reactive oxygen species (ROS) production and oxidative stress, which can lead to mitochondrial dysfunction. Studies have shown that oxidative stress and impaired mitochondrial function can persist in HT patients, contributing to cellular damage and reduced energy production, worsening physical and mental symptoms. Recent studies suggest that understanding the relationship between HT and oxidative stress is essential for HT management. ROS is an important indicator of oxidative stress in biological systems and can lead to the disruption of cellular homeostasis. Mitophagy is of great importance to protect the cell from high levels of ROS, to ensure the beta-oxidation of fatty acids, and to maintain the mitochondrial quality required for oxidative phosphorylation. Thyroid hormones induce mitochondrial mitophagy, and in this process, damaged mitochondria are removed from the cell, preventing increased oxidative stress and apoptosis. This mechanism plays a critical role in maintaining cellular homeostasis and optimal metabolic function [31]. Clinical studies clearly demonstrate that the balance between oxidants and antioxidants is shifted to the oxidative side in patients with HT. Studies in animal models, especially in the NOD.H2h4 mouse model, confirm that ROS accumulation is involved in the onset and progression of autoimmune thyroiditis. Dietary habits and antioxidant supplements may play an essential role in the prevention of autoimmunity [41].

In a study of 124 people, 93 of them were HT patients and 31 were healthy volunteers. In this study, total oxidant status (TOS) and oxidative stress index (OSI) levels were significantly higher in the HT group, while total antioxidant status (TAS), total thiol, and arylesterase levels were lower. Furthermore, a negative correlation between Anti-TPO and TAS and between Anti-TG and total thiol was shown. These findings suggest that oxidative stress is gradually increasing in HT patients, and oxidative balance is impaired with the progression of the disease [42].

The pathogenic role of ROS and oxidative stress observed in the thyroid gland has been similarly reported in the brain. This mechanism may be effective in the development of many neurodegenerative diseases. The brain’s rich lipid content, high energy demand, and limited antioxidant capacity make it vulnerable to high levels of ROS. This explains another reason for the low MCS-12 scores in the HT group, despite long-term LT4 treatment. A link between Hashimoto’s thyroiditis and cognitive dysfunction, changes in consciousness, and neuropsychiatric symptoms has also been reported. A condition described since 1966, Hashimoto’s encephalopathy (HE), points to this association. HE is a rare disease with non-specific clinical manifestations, and its onset can be insidious. In most patients, cognitive function and behavioral/personality disorders are the first symptoms. To differentiate HE from autoimmune encephalopathies and paraneoplastic syndromes, antibody titers against thyroid antigens should be evaluated. Corticosteroid treatment should be considered if antibody titers are high [43,44].

One important feature that distinguishes our study from other studies in this field is the exclusion of patients with additional chronic diseases (hypertension, heart failure, diabetes mellitus, autoimmune thyroid disease, etc.). The other is that the duration of LT4 treatment is at least 3 years.

One of the limitations of our study is the exclusion of sociologic conditions that may affect the SF-12 results, such as marital status, and education level, monthly income of the HT, and control groups participating in the study. Another limitation is the small size of our case sample group. This occurred because we included HT patients who had been regularly followed up in our outpatient clinic for 3 years who were under LT4 treatment. Moreover, they had no other chronic disease that would affect their quality of life. Another limitation is the lack of follow-up data for the HT group. Our inclusion criterion was at least three years of LT4 treatment; however, we did not define and record the upper limit of the follow-up period, which prevented us from calculating the median follow-up period. In future studies, recording the follow-up period in more detail will enable us to obtain more reliable results. Also, about this situation, our study cannot specifically examine the potential effects of the duration of LT4 use. This should be considered as a limitation in the interpretation of the results.

## 5. Conclusions

In this study, the quality of life of euthyroid patients receiving LT4 treatment for HT-induced hypothyroidism for at least 3 years was compared with a control group with similar demographic characteristics using the SF-12 scale. Our study revealed that euthyroid HT patients had lower mental and physical health subs-cores than the control group. Our findings confirm that HT affects women more than men and that the disease is associated with genetic predisposition. Furthermore, we observed that euthyroid HT patients who received long-term LT4 replacement therapy and were regularly followed up had high levels of Anti-TPO and Anti-TG antibodies.

In conclusion, our study suggests that the effective control of hypothyroidism is not sufficient to reduce the adverse effects of HT on patients’ health-related quality of life. Beyond the normalization of hormone levels, comprehensive treatment strategies targeting autoimmune aspects of the disease are needed in the management of HT. This study provides a basis for the development of more effective therapeutic approaches that can improve the quality of life of HT patients.

## Figures and Tables

**Figure 1 jcm-13-03082-f001:**
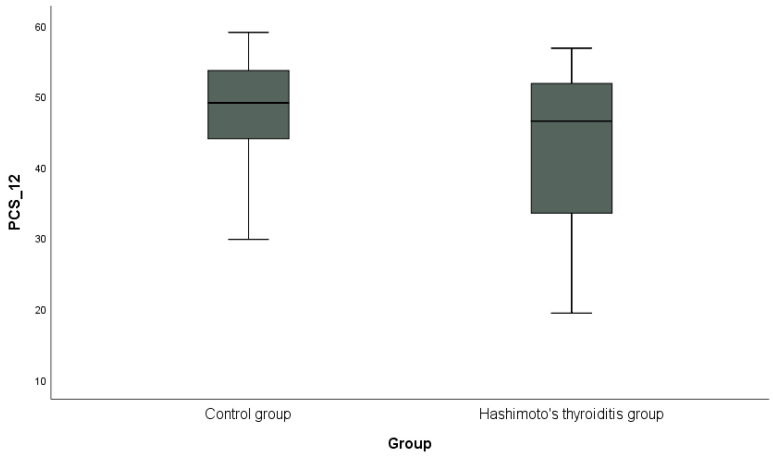
Graphical representation of the physical subscale score (PCS-12) of the SF-12 scale in hypothyroidism and control groups.

**Figure 2 jcm-13-03082-f002:**
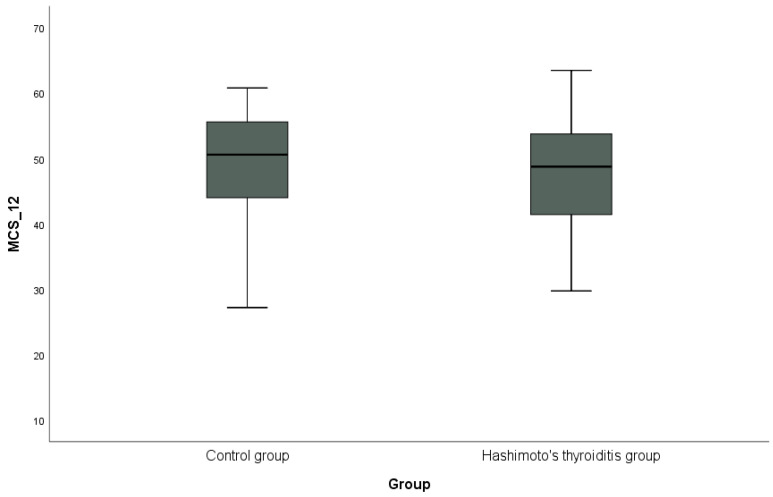
Graphical representation of the mental subscale score (MCS-12) of the SF-12 scale in hypothyroidism and control groups.

**Table 1 jcm-13-03082-t001:** Comparison of demographic and clinical characteristics between groups.

	Hashimoto’s Thyroiditis (n = 44)	Control (n = 44)	*p*
Age, year	49.50 ± 12.09	47.43 ± 9.89	0.136 ^a^
BMI, kg/cm^2^	28.86 ± 5.46	26.62 ± 3.93	0.073 ^a^
Gender, female	35 (79.5)	31 (70.4)	0.325 ^c^
Alcohol, yes	6 (13.6)	6 (13.6)	1.000 ^c^
Alcohol, year	3.52 ± 10.26	2.41 ± 6.59	0.876 ^b^
Smoking, yes	12 (27.3)	15 (34.1)	0.449 ^c^
Cigarettes, pack/year	7.32 ± 13.48	8.79 ± 13.60	0.510 ^b^
Family history, yes	26 (59.1)	12 (27.3)	0.002 ^c^
CAD, yes	10 (22.7)	5 (11.4)	0.156 ^c^
Anti-lipidemic drug use, yes	11 (25.0)	7 (12.6)	0.364 ^c^
Hemoglobin	13.17 ± 1.36	13.61 ± 2.51	0.082 ^a^
TSH	2.92 ± 1.77	1.84 ± 0.85	0.003 ^b^
T3	2.83 ± 0.41	2.97 ± 0.49	0.054 ^b^
T4	1.30 ± 0.28	1.22 ± 0.49	0.021 ^b^
Triglycerides	118.75 ± 48.45	130.52 ± 94.53	0.818 ^b^
Cholesterol	195.54 ± 39.23	197.54 ± 46.37	0.766 ^a^
HDL	50.79 ± 12.25	56.38 ± 24.53	0.851 ^b^
LDL	130.50 ± 33.63	144.85 ± 92.37	0.599 ^b^

Normal values: T3, 2.0–4.4 ng/L; T4, 0.93–1.7 ng/dL; TSH, 0.27–4.20 mIU/L; HDL, >50-mg/dL; hemoglobin, 11.7–15.7 g/dL; triglycerides, 0–150 mg/dL; cholesterol, 0–200 mg/dL; LDL, 0–130 mg/dL; BMI = body mass index, 18.5–24.9 kg/cm^2^; CAD = coronary artery disease; mean ± standard deviation, n (%); ^a^ Student’s *t*-test; ^b^ Mann–Whitney U test; ^c^ Yates chi-squared test.

**Table 2 jcm-13-03082-t002:** Mean levels of Anti-TPO and Anti-TG in the Hashimoto’s thyroiditis group.

Variables	Mean ± SD	Normal Values
Anti-TPO	95.69 ± 163.15	<34 IU/mL
Anti-TG	38.28 ± 55.91	<40 IU/mL

Anti-TPOs = anti-thyroid peroxidase antibodies. Anti-TGs = anti-thyroglobulin antibodies. Mean ± SD = mean ± standard deviation.

**Table 3 jcm-13-03082-t003:** Spearman’s Correlation Analysis Results for the HT Group.

		Age	Anti−TPO	Anti−TG	BMI	TSH	T3	T4	Triglyceride	Cholesterol	HDL	LDL	PCS−12	MCS−12
Age	r	1.000	−0.180	−0.486	0.129	0.039	−0.208	0.052	0.253	0.112	−0.065	0.118	−0.085	0.216
	p		0.243	0.001	0.405	0.803	0.175	0.738	0.097	0.470	0.677	0.444	0.583	0.159
Anti−TPO	r	−0.180	1.000	0.446	−0.050	−0.108	0.311	0.085	0.020	−0.118	0.036	−0.082	−0.043	−0.135
	p	0.243		0.002	0.747	0.487	0.040	0.585	0.897	0.446	0.815	0.599	0.780	0.381
Anti−TG	r	−0.486	0.446	1.000	−0.145	0.061	0.236	−0.144	0.015	0.053	0.185	0.040	0.274	−0.082
	p	0.001	0.002		0.349	0.693	0.122	0.352	0.923	0.734	0.229	0.799	0.072	0.597
BMI	r	0.129	−0.050	−0.145	1.000	−0.079	0.341	−0.164	0.221	−0.081	−0.227	−0.020	−0.275	0.014
	p	0.405	0.747	0.349		0.610	0.024	0.286	0.150	0.602	0.138	0.900	0.071	0.930
TSH	r	0.039	−0.108	0.061	−0.079	1.000	−0.152	−0.556	−0.049	0.110	−0.107	0.072	0.198	0.110
	p	0.803	0.487	0.693	0.610		0.325	0.000	0.751	0.476	0.490	0.642	0.198	0.479
T3	r	−0.208	0.311	0.236	0.341	−0.152	1.000	0.203	−0.027	−0.303	−0.038	−0.288	−0.063	−0.174
	p	0.175	0.040	0.122	0.024	0.325		0.186	0.861	0.046	0.805	0.058	0.685	0.258
T4	r	0.052	0.085	−0.144	−0.164	−0.556	0.203	1.000	−0.037	−0.251	−0.133	−0.228	0.064	−0.110
	p	0.738	0.585	0.352	0.286	0.000	0.186		0.812	0.100	0.388	0.136	0.678	0.477
Triglyceride	r	0.253	0.020	0.015	0.221	−0.049	−0.027	−0.037	1.000	0.188	−0.392	0.160	−0.138	0.160
	p	0.097	0.897	0.923	0.150	0.751	0.861	0.812		0.221	0.008	0.299	0.373	0.301
Cholesterol	r	0.112	−0.118	0.053	−0.081	0.110	−0.303	−0.251	0.188	1.000	0.258	0.945	0.106	0.157
	p	0.470	0.446	0.734	0.602	0.476	0.046	0.100	0.221		0.090	0.000	0.492	0.309
HDL	r	−0.065	0.036	0.185	−0.227	−0.107	−0.038	−0.133	−0.392	0.258	1.000	0.096	0.087	−0.277
	p	0.677	0.815	0.229	0.138	0.490	0.805	0.388	0.008	0.090		0.533	0.576	0.069
LDL	r	0.118	−0.082	0.040	−0.020	0.072	−0.288	−0.228	0.160	0.945	0.096	1.000	0.077	0.146
	p	0.444	0.599	0.799	0.900	0.642	0.058	0.136	0.299	0.000	0.533		0.620	0.343
PCS−12	r	−0.085	−0.043	0.274	−0.275	0.198	−0.063	0.064	−0.138	0.106	0.087	0.077	1.000	0.302
	p	0.583	0.780	0.072	0.071	0.198	0.685	0.678	0.373	0.492	0.576	0.620		0.047
MCS−12	r	0.216	−0.135	−0.082	0.014	0.110	−0.174	−0.110	0.160	0.157	−0.277	0.146	0.302	1.000
	p	0.159	0.381	0.597	0.930	0.479	0.258	0.477	0.301	0.309	0.069	0.343	0.047	

r: Spearman’s correlation coefficient. Values range between +1 and −1; +1 indicates a perfect positive correlation, −1 indicates a perfect negative correlation, and 0 indicates no correlation; p: statistical significance value. *p* < 0.05 indicates that the correlation is significant at the 95% confidence level; *p* < 0.01 indicates significance at the 99% confidence level; N: sample size, which is 44 in this study.

**Table 4 jcm-13-03082-t004:** Comparison of physical (PCS-12) and mental (MCS-12) subscale scores of the SF-12 scale between the groups.

Group	Hashimoto’s Thyroiditis (n = 44)	Control (n = 44)	*p* *
PCS-12 (Mean ± SD)	42.79 ± 11.13	48.11 ± 7.03	0.049
MCS-12 (Mean ± SD)	46.84 ± 9.76	48.75 ± 9.29	0.346

Mean ± standard deviation. * Mann–Whitney U test.

## Data Availability

Our study data contain personal information of patients, and therefore, are not available for sharing due to the ‘Personal Data Protection Law’ and ethical reasons.

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
