# Peer review of "Evaluation of Health-Related Quality of Life in Patients with Euthyroid Hashimoto’s Thyroiditis under Long-Term Levothyroxine Therapy: A Prospective Case-Control Study"

_jcm, 2024, doi:10.3390/jcm13113082_

Round 1

Reviewer 1 Report

Comments and Suggestions for Authors

Reviewer

Initial comments

         This paper is important because the prevalence of thyroid diseases is rapidly increasing (1), and autoimmune thyroid disease (AITD) is one of the most common, being observed in approximately 5% of individuals (2).

In iodine sufficient regions, the major cause of Primary Hypothyroidism  is AITD (10).

1. Lombardi FA, Fiore E, TonaccheraM,  Antonangeli L, Rago T, Frigeri M, et al.The effect of voluntary Iodine Prophylaxis in a Small Rural Community: the Pescopagano Survey 15 Years Later. J Clin Endocrinol Metab( 2013) 98 (3):1031-1039. doi.org/10.1210/jc.2012-2960

2.Fröhlich E, Wahl R. Thyroid Autoimmunity: Role of Anti-thyroid Antibodies in Thyroid and Extra-Thyroidal Diseases. Front Immunol. 2017; 8:521.doi: 10.3389/fimmu.2017.00521

10. Dayan, C.M.; Daniels, G.H. Chronic autoimmune thyroiditis. N. Engl. J. Med. 1996, 335, 99–107

        Thyroid hormones regulate normal growth, development, and various homeostatic functions, particularly the production of energy and heat.

        The patient medicated with levothyroxine alone, anti-thyroperoxidase (A-TPO) and anti-thyroglobulin (A-Tg) antibodies remain elevated, and there is no treatment for Hashımoto's Thyroiditis (HT) and only for hypothyroidism, opening up a new field for research.

   Thus, this paper is very important and can bring a great contribution to patients with HT and hypothyroidism,

Title: Evaluatıon of Health-Related Qualıty of Lıfe in Patıents wıth Euthyroıd Hashımoto's Thyroiditis under Long-Term Levothyroxıne Therapy: A Case-Control Study

Comment:

It is suitable

Abstract:

  Comment:

 It is suitable

1. Introduction  

 Comment:

Lines 35,36,37….. Hashimoto's thyroiditis (HT) is the most common autoimmune thyroid disease characterized by thyroid gland enlargement with atrophy secondary to autoimmune-induced inflammation.

Please, what about the reference?

Segundo Amino et al. Hashimoto's thyroiditis (HT) was considered when a goiter was present (13).

13. Akamizu T, Amino N, De Groot LJ. Hashimoto’s Thyroiditis.Thyroid Disease Manager.2008.http://www.thyroidmanager.org/Chapter8/8-contents.htm

Line 58,59….HT is diagnosed with elevated thyroid stimulating hormone (TSH) levels and positive autoantibody tests.

TSH is a laboratory test to verify thyroid classification and function and not for the diagnosis of HT.

Please correct this paragraph,

          Chronic autoimmune thyroiditis was diagnosed in patients who were positive for antibodies against thyroglobulin (A-Tg) and thyroperoxidase (A-TPO) and who had a heterogeneous texture and marked hypoechogenicity on ultrasound of the thyroid gland. Hashimoto's thyroiditis (HT) was considered when a goiter was present (13).

13. Akamizu T, Amino N, De Groot LJ. Hashimoto’s Thyroiditis.Thyroid Disease Manager.2008.http://www.thyroidmanager.org/Chapter8/8-contents.htm

Line 60... Currently, levothyroxine (LT4) is widely used in the treatment of HT   ????????????????

Or is it hypothyroidism that we treat with sadistic levothyroxine?

Please correct.

Lines 61, 62,63…In addition, LT4  helps maintain stable levels of T3 hormone in the body by converting T4 to triiodothyro nine (T3) in peripheral tissues.

Please check the literature, as it is necessary to cite the deiodinases here.

Lines 65,66,67…The general consensus is that all patients with a serum TSH level of 10 mU/L should be treated. The goal of treatment is to achieve the set TSH target (0.4-4.0 mIU/L).

Please cite reference

Lines 67,68,69,70…However, when adjusting treatment  according to the TSH level, it is essential to remember that TSH release increases with age,  obesity, and smoking may slightly increase TSH levels, pregnancy may decrease TSH levels, and TSH release has a daily rhythm.

Please cite reference

2. Materials and Methods 

2.1. Study Design and Participants

Comment:

It is suitable

2.2. SF-12 Questionnaire Form

Comment:

It is suitable

2.3. Ethical Approval

Comment:

It is suitable

2.4. Statistical Analysis

Comment:

It is suitable

3. Results

Comment:

It is suitable

4. Discussion

Comment:

Lines 227,228,229,230…It is known that metabolic syndrome is more common in patients with HT. This condition leads to chronic comorbidities (hypertension, diabetes mellitus, coronary artery disease) and, thus, to a decrease in the quality of life of the patients. This can be controlled  with an appropriate dose of LT4 replacement [1].

Lines 230,231…The findings in our study are consistent with the literature.

Please cite reference

5. Conclusions

Comment:

Lines  299,300,301,302,303,304….,One of the limitations of our study is the exclusion of sociologic conditions that may  affect the SF-12 results, such as marital status, education level, monthly income of the HT,  and control groups participating in the study. Another limitation is the small size of our  case sample group. This occurred because we included HT patients who had been regularly followed up in our outpatient clinic for 3 years, were under LT4 treatment, and had  no other chronic disease that would affect quality of life.

These phrases could be in the discussion.

And the conclusion follows the abstract.

Abstract:

Lines 24,25,26,27……..Conclusions :The  study indicates that Hashimoto's thyroiditis adversely impacts the quality of life, despite long-term  therapy and low antibody levels. Evaluating quality of life is crucial in the treatment planning for these patients.

References

Comment:

There are 23 references

I think the paper of Chaker et al could be cited.

      According to Chaker et al. (5) the definition of hypothyroidism is based on statistical reference intervals of relevant biochemical parameters due to the great variation in clinical presentation and the general lack of specificity of symptoms, and this topic is increasingly debated.

5. Chaker L, Bianco AC, Jonklaas J, Peeters RP. Hypothyroidism. Lancet (2017) 390:1550-62. doi: 10.1016/S0140-6736(17)30703-1

Why aren't these references more complete with the names of the other authors?

6. K. H. Winther et al., Disease-specific as well as generic quality of life is widely impacted in autoimmune hypothyroidism and 333 improves during the first six months of levothyroxine therapy, PloS one, 2016; vol. 11, p. e0156925. https://doi.org/10.1371/jour-334 nal.pone.0156925 [Google scholar] 335

7. O. Okosieme et al., Management of primary hypothyroidism: statement by the British Thyroid Association Executive Commit-336 tee, Clin. Endocrinol., 2016; vol. 84, pp. 799-808. https://doi.org/10.1111/cen.12824 [Google scholar] 337

11. B. Gandek et al., Cross-validation of item selection and scoring for the SF-12 Health Survey in nine countries: results from the 346 IQOLA Project, J. Clin. Epidemiol., 1998; vol. 51, pp. 1171-1178. https://doi.org/10.1016/S0895-4356(98)00109-7 [Google scholar]

20. P. Vigário et al., Perceived health status of women with overt and subclinical hypothyroidism, Med Princ Pract., 2009; vol. 18, 364 pp. 317-322. https://doi.org/10.1159/000215731 [Google scholar] 365

21. C. B. Larsen et al., Severity of hypothyroidism is inversely associated with impaired quality of life in patients referred to an 366 endocrine clinic, J. Thyroid Res, 2023; vol. 16, p. 37. https://doi.org/10.1186/s13044-023-00178-0

23. M. M. Yalcin et al., "Is thyroid autoimmunity itself associated with psychological well-being in euthyroid Hashimoto’s thyroid-370 itis?, Endocr. J., 2017; vol. 64, pp. 425-429. https://doi.org/10.1507/endocrj.EJ16-0418 [Google scholar]

Thank you

Reviewer 2 Report

Comments and Suggestions for Authors

In this case-control study, the authors investigated the quality of life of euthyroid Hashimoto's thyroiditis patients receiving levothyroxine therapy. Despite its significant scientific merits, I have several concerns that should be covered before considering publication.

  1. First of all, the title contains a few dotless “i” letters, which is not acceptable when writing in English. Please amend.
  2. In introduction:
    • Line 35: Abbreviate Hashimoto thyroiditis upon first mention and not the second mention (i.e., before sentence end).
    • Line 41: Define TSH upon first mention.
    • Line 58: Keep only the abbreviation and don’t overstate the definition.
    • Line 62: Define T3 on first mention.
  1. In Material and Methods, line 129-130:
    • Was this study prospective or retrospective in nature?
    • Please transfer the last sentence to statistical analysis “subheading”
    • Please discuss in detail the parameters collected for HT and control subgroups.

  2. In results:
    • What was the median duration of follow-up for HT subgroup?
    • What was the median duration of levothyroxine intake for HT subgroup and why this was not implemented in the comparison (i.e., in Table 1)
    • Were there any statistical difference when exploring mean anti-TPO and anti-TG levels between HT and control subgroups?
  3. All tables have major style errors and lacks defining the adopted abbreviation in table footnotes. For example, what does VKI stand for?
  4. Current analyzed results only provide a generalized perspective and lack in-depth evaluation of HT subgroup for significant correlation. Authors may apply subgroup correlation analysis (using Spearman's test) to examine and unveil any significant correlation of collected factors in a new subheading and summarized table.
  5. Please start the discussion by showing your study strengths.
  6. The current supplied conclusion belongs to the study limitation section, which is often shared at the end of the discussion. The conclusion section should typically highlight the obtained results in a summarized manner. Please rewrite the conclusion section and move the current written sentences to the end of the discussion.
  7. Finally, I would suggest letting a native English speaker read over this article to improve the fluency and style of this manuscript.
Comments on the Quality of English Language

I would suggest letting a native English speaker read over this article to improve the fluency and style of this manuscript.

Reviewer 3 Report

Comments and Suggestions for Authors Here are my detailed comments regarding Manuscript ID: jcm-2995923:   

In this study, the authors compared the health-related quality of life scale (SF-12) of 44 euthyroid patients with Hashimoto's thyroiditis, who had been on LT4 therapy for at least 3 years with 44 healthy volunteers with equivalent demographic characteristics who did not have chronic diseases that reduce the quality of life.

The study design is elegant, the methods and statistics are good, and the discussion is well-written, despite the small size of the case sample group—a point the authors make several times in the text. The fact that this study only included patients with isolated hypothyroidism (without any other chronic conditions) who received LT4 treatment for at least three years, is another one of its key advantages. This implies that the hormone levels in the study group were steady, which emphasizes the significance of the study's conclusions. Still, patients on LT4 had significantly higher TSH and T4, and borderline significant (p=0.054) lower T3 values. Although the authors explain these data in lines 240–250, I believe that there should be more information provided on the possibility of brain hypothyroidism in levothyroxine users. The reference below could help with this issue:

Hegedüs, L., Bianco, A.C., Jonklaas, J., Pearce, S.H., Weetman, A.P., Perros, P. Primary hypothyroidism and quality of life. Nat Rev Endocrinol 2022; 18, 230–242

Regarding my previous remarks, I would conclude that patients receiving LT4 treatment for primary hypothyreosis due to Hashimoto thyroiditis had a lower quality of life than healthy volunteers. This is not the authors' conclusion, which is only explicitly expressed in the abstract:“ The study indicates that Hashimoto's thyroiditis adversely impacts the quality of life, despite long-term therapy and low antibody levels.”. Since the medication can't precisely mimic our physiology, it is unknown at this point whether the treatment makes these people feel unwell, whether the autoantibodies are the cause or both.

And finally, the authors should emphasize their conclusion in the Conclusion section of the manuscript. In its current form, only the study limitations were emphasized.

Reviewer 4 Report

Comments and Suggestions for Authors

- Elaborate on the selection criteria for control participants (lines 109-111) to ensure clarity on how they match the patient group beyond just the non-presence of thyroid disease. This is critical for readers to assess the study's internal validity.

- The explanation of the statistical methods used (lines 149-165) is succinct, yet it could be improved by specifying why certain tests were chosen over others based on the data distribution.

-Would add prevalence of subclinical hypothyroidism in line 45 as well

- Do the patients with HT have other autoimmune conditions? If they have 1 AI condition, they are more likely to have another AI condition. This could potentially explain the difference in scores. If not this should be added to the limitations.

-Lines 67-70 – please rephrase to make it more clear to the readers

-Line 108-115 – repetition

-              Since most endocrinologists are not aware of the questionnaire, I would add it in the supplementary material

- The limitations mentioned are good (lines 299-304); however, adding potential biases or the study's generalizability could further clarify for the reader the contexts in which the findings are most applicable.

-              Authors only compare their results with 1-2 publications in the past. I would recommend them comparing their results with more data that has been published and highlight how their results differ from the literature that is already published.

- Ensure that terms like "euthyroid Hashimoto's thyroiditis" are consistently used throughout the manuscript to avoid confusion (lines 11 and 211).

-Several minor grammatical errors and awkward phrasings could be corrected to improve readability. For instance, the sentence structure in lines 150-152 is convoluted and could be simplified for clarity. All tables have , instead decimal points

- In table 1- what do VKI and KAH stand for?

-              Table 2 needs normal values as well

- Conclusions: Strengthen the conclusion by succinctly summarizing the study's impact on clinical practices or future research directions, particularly relating to quality of life assessments in Hashimoto's thyroiditis patients (line 298).

Comments on the Quality of English Language

As above 

Round 2

Reviewer 2 Report

Comments and Suggestions for Authors

Adequate attention. Many thanks

Comments on the Quality of English Language

Adequate attention. Many thanks

Reviewer 4 Report

Comments and Suggestions for Authors

Authors have done a good job editing the manuscript and incorporating the feedback 

Comments on the Quality of English Language

Improved 
